# Miniaturized Compact Reconfigurable Half-Mode SIW Phase Shifter with PIN Diodes

Franky Dakam Wappi, Bilel Mnasri, Alireza Ghayekhloo, Larbi Talbi and Halim Boutayeb *

Department of Computer Science and Engineering, University of Quebec in Outaouais (UQO), Gatineau, QC J8Y 3G5, Canada
* Correspondence: halim.boutayeb@uqo.ca; Tel.: +1-800-567-1283 (ext. 2908)

**Abstract:** In this work, a novel electrically reconfigurable phase shifter based on a half-mode substrate integrated waveguide (HM-SIW) is proposed. SIW is a guided transmission line topology, and by using half-mode excitation, a smaller size can be achieved. Phase shifters are electronic devices that change the phase of transmission for a wide range of applications, including inverse scattering and sensing. The tunability of PIN diodes is applied here to achieve a reconfigurable design. The proposed single-layer structure does not require extra wiring layers for the bias circuit on the suggested printed circuit board. Its principle consists in the integration, in the HM-SIW, of three parallel lines, each connecting the edge of the HM-SIW and linked to a PIN diode and a radial stub. Here we present the results of measurements for a frequency band from 4.5 to 7 GHz that demonstrate how the experiment agrees with simulations. Insertion loss was less than −10 dB, and port coupling was less than −2 dB for both simulation and measurement solutions. The proposed half-mode structure is around half the size of a typical SIW line. With the proposed design, the seven states of the PIN diodes can be validated (ON and OFF), with a wide band adaptation and a relatively constant phase difference across a broad frequency range (44%). A key benefit of the proposed design for a microwave component is the reduction of extra biasing layers for the PIN diodes. This is in addition to the reduced size of the transmission line compared to a commercial SIW. In the annexed section, simulation software is used for a more comprehensive analysis involving more phase shift values and parametric studies.

**Keywords:** electronic phase shifter; half-mode; PIN diodes; miniaturized size; reconfigurable microwave device; SIW





## 1. Introduction

The phase shifter is a key microwave circuit in modern communication, especially when designing a phase array system [1–4]. In multidisciplinary antenna devices, a phase shifter can be used to feed both the transmit and receive sections [5–8]. However, a phase shifter is a more practical method of identifying information due to the low amount of power and linearity features at reception [9]. Currently, phase shifters are mainly applied to the electronic beam control of an antenna in a phased array system [10–14]. This is to improve the quality of the coverage area in transmission as well as in high-speed reception. Other applications are cavity filters, power dividers, phase modulators, high power devices, and radars [15]. Phase shifters can be designed using passive or active elements.

According to the literature, there are different types of phase shifters [16,17]. We can create phase shifts based on transmission lines to create an arbitrary degree of phase shift in a frequency band [16]. This technique has been applied to Schiffman-coupled differential transmission lines. Furthermore, high-power handling waveguides can generate a phase step from a multi-step ridge section along a shift guided line [17]. Moreover, filtering mechanisms undergo different transmission line comparisons for phase shifting. With this mechanism, it is possible to provide constant phase delays [18]. A dual-band phase shifter

has been proposed by comparing two different transmission lines for two distinct phase shifts [19]. A more complex structure has been applied to design a tunable filter phase shifter at 1 GHz frequency with two power dividers and two reconfigurable filter couplers [20]. Additionally, substrate-integrated waveguides (SIW), as a new generation of transmission lines [21–23], can be configured for phase-shifting microwave networks [24–26]. Added metal posts to change SIW resonators [24], slots on the upper surface of the SIW [25], high dielectric constant perturbation [26,27], fixed cross-sectional elements [28], and coupled slots [29] are some of the models presented to develop a SIW phase shifter.

Many of these methods generate a fixed phase difference between two points in the microwave component. However, it is highly desirable to create a modular or reconfigurable phase shifter device. In particular, a compact and miniaturized element is required that can be easily integrated with other microwave-packaged components. By using a diode and its related states, one can tune the constant phase shifter structure as studied in [30,31]. The PIN diodes can interact with the SIW line to change the required phase difference while the signal travels through the line. Integration of diodes in an electromagnetic device is not straightforward and requires innovative aspects regarding diode state change. An example of a technique presented in a patent from 2022 involves eliminating the need for additional biasing lines for PIN diodes used in beam switching applications [32]. This algorithm is suggested to generate a phase shifter microwave component in this novel progressive work. The intervention load parameters may also be modified by using a varactor and capacitor in addition to the PIN diodes [33–35]. However, a PIN diode could provide a more stable and quicker response to bias and RF sources.

With a tapered microstrip choke and resistor in one united layer, radio frequency (RF) signals and DC current are isolated [36]. In order to separate two energy sources and their functionality, this step must be taken. In addition to the required phase shift and simple one-layer design, it is necessary to reduce the total surface of a compact microwave device. As a result, we propose a half-mode SIW to realize a novel practical electronic device. The half-mode SIW is smaller and performs similarly to a typical SIW phase shifter. The proposed structure can be applied to miniaturized RF circuits to generate desirable phase shifts. A similar dielectric constant must exist to compare the width and size of a phase shifter component to determine the amount of miniaturization. Using the concept of magnetic wall, this paper proposes the steps for designing a new half-mode substrate integrated waveguide (HM-SIW), which presents all the advantages of a SIW but with a size reduced by half [37–39]. Thus, three microstrip lines are connected to three PIN diodes, which are connected at each end to a radial stub. This circuit generates an electrically controllable phase shift between the two ports of a HM-SIW thanks to the eight active states of the diodes associated with the radial stubs. To validate the proposed design, we used several simulation techniques and a concrete measurement step. The length and width of the microstrip lines and their location on the HM-SIW are optimized to have an approximately constant phase shift over the considered frequency range. The rest of the paper is organized as follows: Section 2 presents the design of the HM-SIW and the proposed HM-SIW phase shifter component. Simulation results and fabricated circuit tests for optimized structures are presented in Section 3. Moreover, concluding remarks are given in Section 4. In addition, two useful Appendix A and Appendix B that investigate a high phase shift feature and some parametric studies with the same proposed technology from simulation software have been prepared.

## 2. Design of the HM-SIW Phase Shifter

The proposed HM-SIW consists of a one-layer substrate, and the background is covered with copper all around. The substrate material is Rogers RT/duroid 5880 with 2.2 relative permittivity ($\varepsilon_r$), 0.508 mm thickness, and 0.0009 loss tangent. The first step in a reconfigurable compact modular phase shifter design is to create a proper transmission line. This transfers signals within the frequency band with low dissipation.

### 2.1. Transmission Line Topology

Figure 1 illustrates the geometric parameters of a HM-SIW structure that transfers a desired signal. The width of the transmission line can be reduced when using a half-mode SIW, but on the other hand, most of the SIW features are still maintained. Instead of vias in both corners, one edge will be an open aperture.

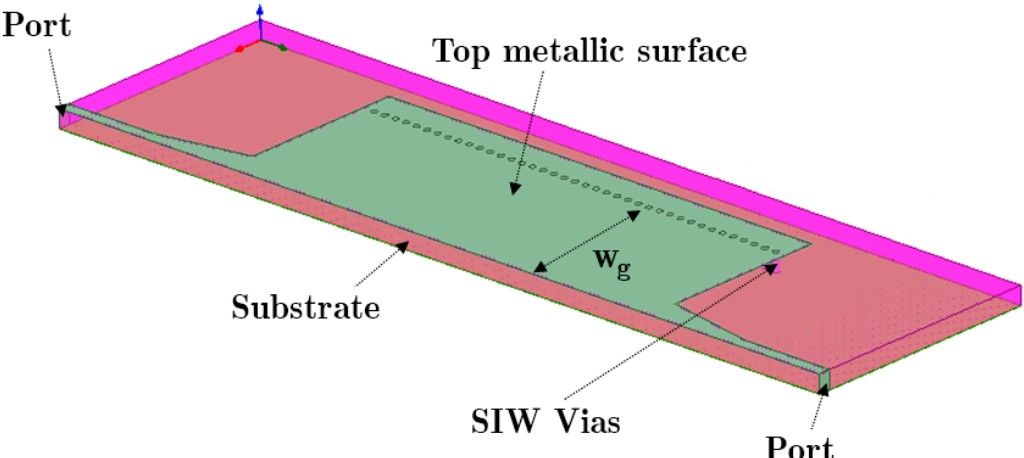

**Figure 1.** Schematic of the proposed HM-SIW structure from simulation software.

Dividing a SIW transmission line in half creates the HM-SIW. On the open aperture plane, the magnetic field is nearly parallel, which creates an equivalent perfect magnetic conductor. This topology can handle a lower amount of signal transmission power and requires a smaller frequency wavelength compared to the thickness of the substrate. Both of the aforementioned facts are acceptable for electronically controlled phase shifter designs and the desired operation frequency of 4~8 GHz.

In addition, the port widths are configured to transmit 50 Ω from microstrip line theory [40]. This configuration is being continued to prepare a constant matching feature to be used with the HM-SIW structure. A typical microstrip line has been used to feed the proposed structure. For adequate impedance matching, the microstrip line segment connecting the SIW has been tapered. We refer to the employed structure as the microstrip to SIW transition [41].

Starting from the cutoff frequency and exploiting the operating conditions of a SIW line, width ($W_g$), pitch, or distance between the vias ($p$), and via diameter ($d$) can be formed. Equations (1)–(9) describe the conditions that govern SIW operation in dominant mode [42]. Then, by exploiting the phenomenon of the magnetic wall, we can define HM-SIW [43].

$$\lambda_g = \frac{c_0}{\sqrt{\epsilon_r}} \, , \tag{1}$$

$$\frac{\lambda}{2} \; < \; W_{eff} \; < \; \lambda, \tag{2}$$

$$p \; < \; d(\text{Physically realizable condition of vias.}), \tag{3}$$

$$\frac{p}{\lambda_g} \; < \; 0.25(\text{avoiding bandgap}), \tag{4}$$

$$d < \frac{\lambda_g}{5}, \text{ and } p \leq 2d \text{ (reducing radiation loss)}, \tag{5}$$

$$\frac{p}{\lambda_c} > 0.05 \; (\text{avoiding over perforation}), \tag{6}$$

$$W_{eff} = W_g - 1.08\frac{d^2}{p} + 0.1\frac{d^2}{W_g} \, , \tag{7}$$

$$\frac{p}{d} < 3, \text{ and } \frac{d}{W_g} < 0.2 \text{ (accurate range)}, \tag{8}$$

$$L_{eff} = L - 1.08\frac{d^2}{p} + 0.1\frac{d^2}{L}, \tag{9}$$

where $c_0$, $\lambda_g$, $L_{eff}$, and $W_{eff}$ are, respectively, the speed of light, guided wavelength, effective length, and width [44]. Equations (1) and (2) are applied to calculate the transmission line geometry for 4 GHz when modeling SIW with a dielectric-filled rectangular waveguide. Then, the $W_{eff}$ is split in half for a half-mode SIW. Furthermore, the diameter of the via (d) and the distance between the vias (*p*) are set up with Equations (3)–(9) and the fabrication limits. The diameter of the via holes is $d = 1.016$ mm, and the distance between two via centers is $p = 1.5$ mm. Each via is 0.492 mm away from the metal edge, and $W_g = 15$ mm.

### 2.2. Phase Shifting Topology

A transmission line has its own fixed-phase feature. To change the phase of the transmitted signal, it is possible to integrate parallel posts with the fixed HM-SIW. As shown in Figure 2, three different microstrip lines with lengths P1, P2, and P3 are connected to the device. Three PIN diodes are added to the lines to change the load of the new integrated lines. Additionally, a 115 Ω resistor and a radial stub end are provided with L5 length and θ angle. The resistor, radial stub, and meander line after the PIN diode function as a DC/RF blockage. To reduce the requirement for additional technology in the design, this isolation technique is implemented on the same layer as the RF phase shifter.

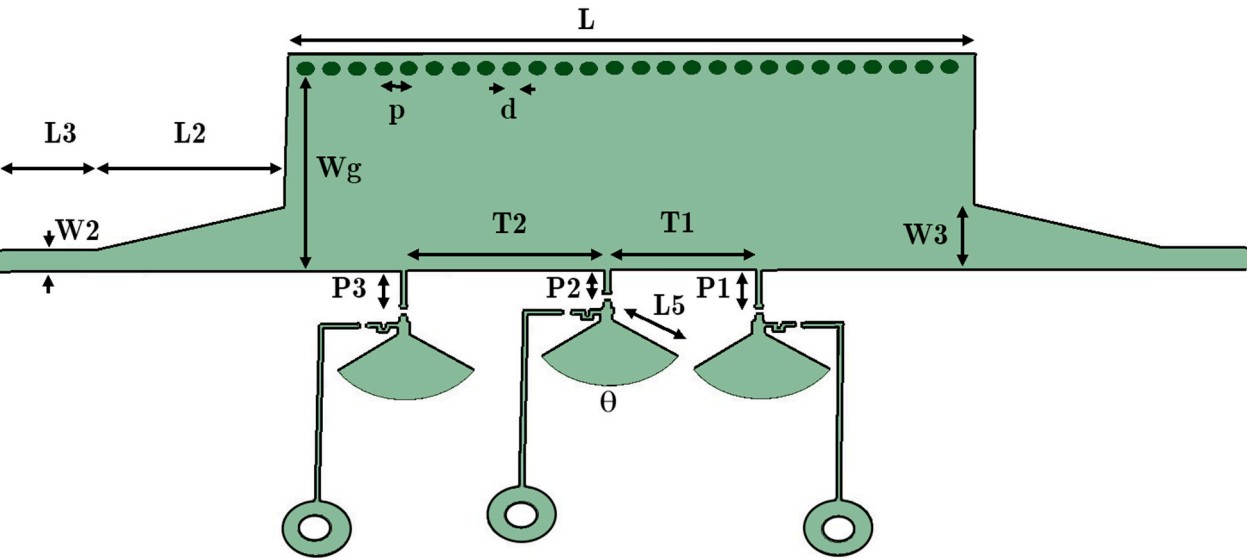

**Figure 2.** Structure of the phase shifter with the HM-SIW including diode posts and RF/DC blocking system from simulation software (T1, T2, L2, L3, W2, W3, Wg, L, P1, P2, and P3 parameters are, respectively, 11.55, 8.55, 11, 12, 1.59, 4.7, 14.5, 40, 2.8, 1.8, and 2.8 mm).

Depending on the DC bias support of PIN diodes, their situation may change (voltage > 0.8 V). To block interference between the two sources, the DC/RF isolating technique is implemented (Figure 3 illustrates this technique). To generate a broad resonant RF short circuit, a radial stub is used (L5 = 4.61 mm and θ ≈ 110°). It can provide a decoupling network between one source and other components. Furthermore, PIN diodes must be modeled in order to be exploited in numerical software. The electronic circuit model of a PIN diode is shown in Figure 4. A parallel RLC circuit is applied instead of a PIN diode element for the HM-SIW, including both diode states. During the ON state, it can be a parallel resistor or an inductance. On the other hand, it is a parallel capacitor and resistor in the OFF state. The diode is modeled as a low-value resistor in the ON state.

In contrast, it is a high-value resistor when it is in the OFF state. A parallel RLC circuit model is exploited because some simulation software, such as ANSYS HFSS, is only able to consider parallel RLC elements.

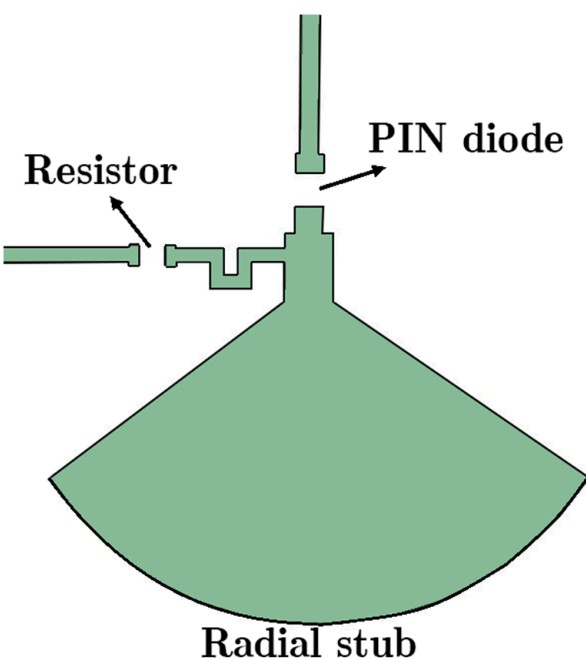

**Figure 3.** Specification of the PIN diode placement in the reconfigurable phase shifter design. The resistor is one element to isolate RF signals with DC bias.

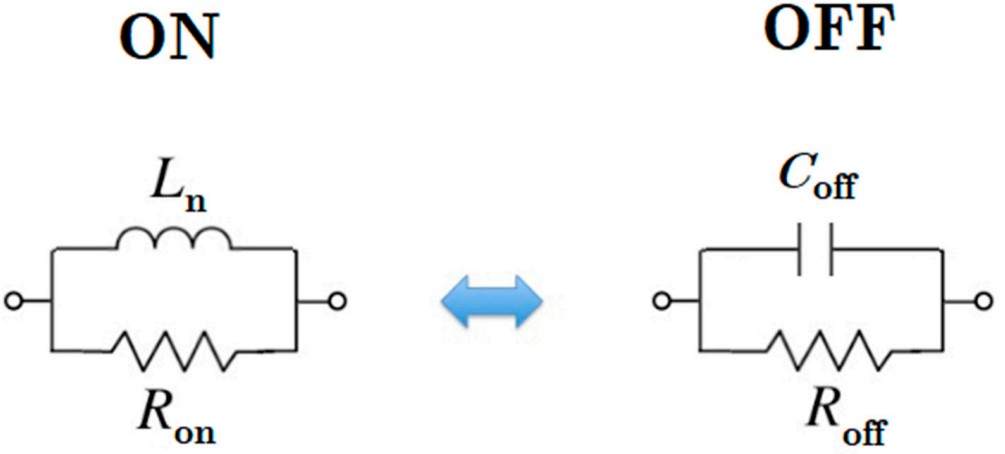

**Figure 4.** Equivalent circuit model of a PIN diode for two different states. Using a parallel arrangement for both states, the ON state is derived from a series-to-parallel transformation. ($L_n$ = 0.209 nH, $R_{on}$ = 33.3 Ω, $R_{off}$ = 30 kΩ, and $C_{off}$ = 0.1338 pF).

The PIN diode is MPP4201 from Microsemi, and the resistor is from KOA Speer Electronics. The required voltage for changing the diode situation to an ON state was 0.8 V. For the linear performance of the diode, a stable DC voltage source with the desired switch is used.

### 2.3. Working Principle

This subsection provides a qualitative perspective on the principle of structure. One method to analyze the structure is to hypothesize HM-SIW as a transmission line [45]. Despite being a high-pass filter by considering the cut-off frequency, the HM-SIW's primary mode reduces signal transmission to a band-pass. The equivalent circuit element

of a high-pass filter with one parallel stub is modeled as a T-network high-pass filter circuit in Figure 5 [24,30]. By adding inductance to the parallel branch, it is possible to increase the phase shift when compared with the reference value. This procedure is applied in the schematic of Figure 6. The lumped element values for the high-pass filter are from an equivalent value for a Butterworth model [46]. Then diode post values can tune both the phase and amplitude values of the reference HM-SIW. This principle is applied here for phase shifting applications with parametric studies, simulation software, and concrete measurement.

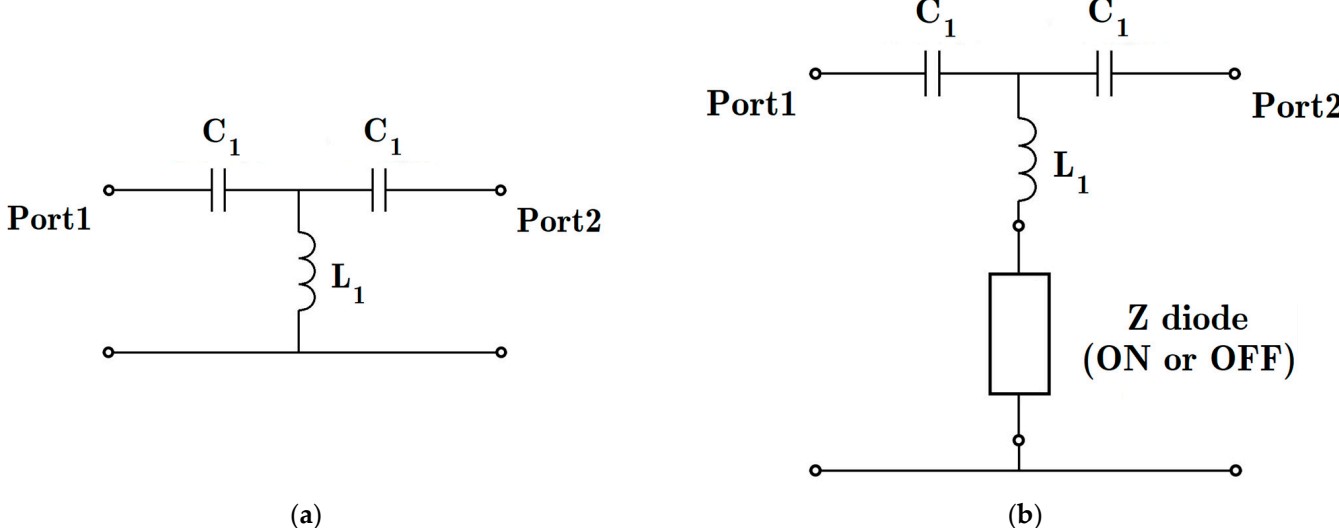

(a)　　　　　　　　　　　(b)

**Figure 5.** Qualitative circuit model analysis of (**a**) a high-pass filter equaling the HM-SIW guided transmission line and (**b**) the effect of parallel impedance posts on the SIW topology ($C_1$ = 1.061 pF and $L_1$ = 1.326 nH). Port locations are shown in Figure 1. Furthermore, Figure 4 displays the values for the diode impedance (Z diode).

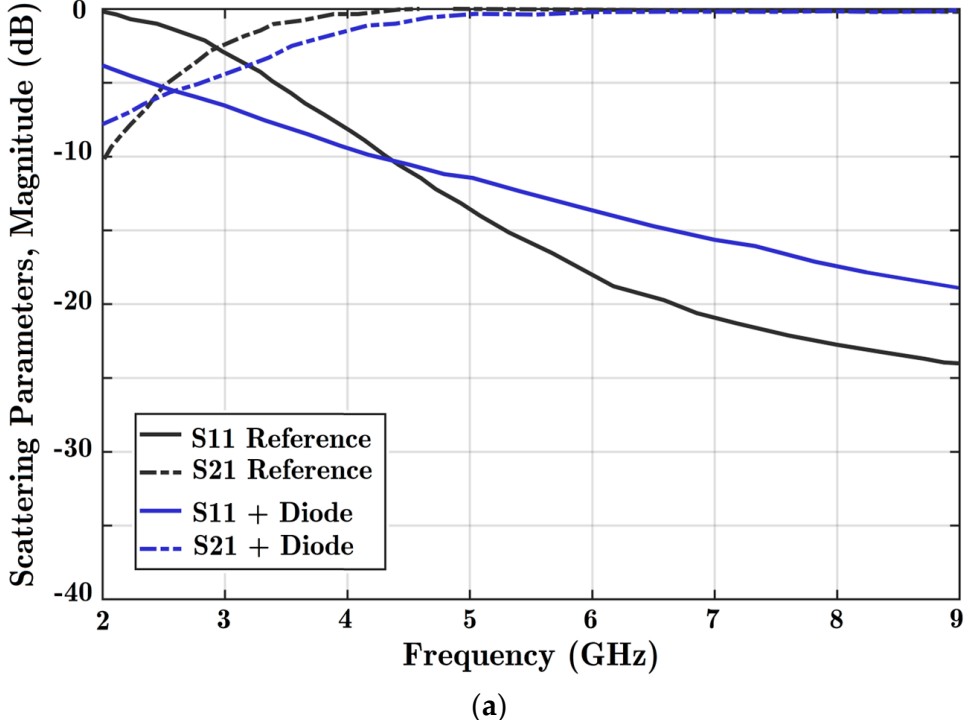

(a)

**Figure 6.** *Cont.*

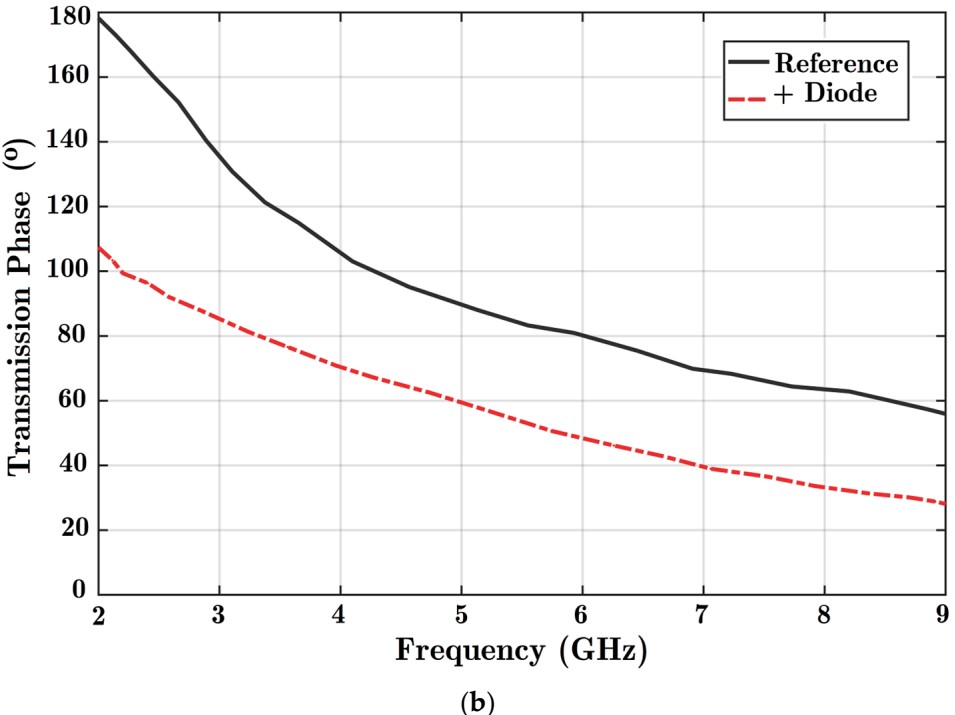

**(b)**

**Figure 6.** Analysis of the effects of adding parallel stubs to the transmission line. Charts of (**a**) scattering parameter magnitudes and (**b**) transmission phase when the HM-SIW is alone (reference) and by adding a parallel diode (ON state) are prepared.

When looking at Figure 2, the added length of lines connecting to the PIN diodes presents a constant phase shift for every branch. Additionally, they provide enough space for a diode soldering pad. The location of PIN diode stubs is tuned with full wave simulation, considering fabrication constraints and achieving a sensible phase shift for a smaller total length.

## 3. Results and Discussion of Varied Solutions

The proposed structure was designed in numerical full wave simulation software according to the geometry of Figure 2. In light of the scattering parameters, the geometry was tuned to provide suitable performance across the frequency range. The background surface is applied with a perfect electric conductor as the boundary condition. Other directions are chosen as open added-space boundaries. Two ideal waveguide ports with larger areas are added to the two microstrip lines at the corners. The frequency of analysis in the simulation software is 2 to 9 GHz. Various simulation software, including CST, ANSYS HFSS, and a circuit analyzer (ADS), is used to verify the structure before a concrete measurement study.

To further evaluate a HM-SIW phase shifter microwave component, the proposed structure is fabricated as shown in Figure 7. Two SMA connectors are soldered to the input and output ports. In the figure, the metallic surface of the HM-SIW is covered with copper and solder paste. A portable network analyzer from Keysight (N9950B) is connected to investigate the scattering parameters for two ports, including reflection and coupling values both in dB and phase units. Before performing the scattering parameter measurements, a mechanical 2-port calibration is conducted (based on a broadband SOLT kit). PIN diodes and resistors are soldered to the board. Then, a DC source is employed to change the diode situation using key control. For better soldering on the same SIW surface, three extra arms connected in one direction to the diode are provided. On the opposite side of the background, the diodes are connected to a DC source in the other direction.

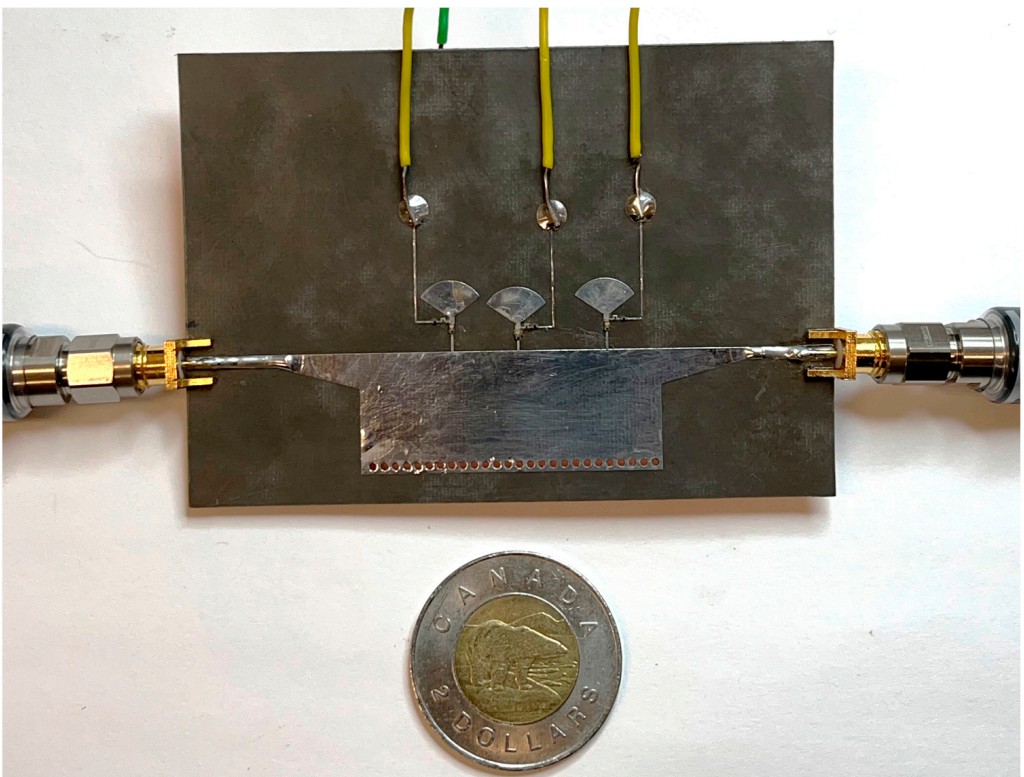

**Figure 7.** A fabricated HM-SIW phase shifter that can be easily integrated with other microwave components due to its compact, modular, and miniaturized features.

There are three diodes in the design; therefore, eight different formations of diode states can be explored. The first state, 000, shows that all three diodes are OFF. Moreover, 111 indicates that all three diodes are in the ON state. The magnitude of scattering parameters in dB is obtained in these two situations from varied solutions. In Figure 8, the scattering coefficients for both ports are acceptable from 4.5 to 7 GHz for both scenarios. Based on the different solutions, the reflection coefficients of both ports were below −10 dB, and the coupling parameter was below −2 dB considering the assigned frequency band. The coupling parameter was not acceptable to work at frequencies exceeding 7 GHz in the experiment because of loss in the connectors.

Once proper wave transmission has been carried out in the allocated frequency band, both solutions are examined to determine the phase of transmission. Since three PIN diodes are integrated in the transmission line, there would be eight different situations for diodes. State 000 assigns the first and reference situations. Then, by changing the states of the diodes one by one, the transmission phase variation is analyzed. The phase variation can be controlled digitally with this reconfigurable phase shifter. Figure 9 presents the phase difference resulting from normalizing different diode states to the reference 000 formation.

A maximum 55° phase shift is achieved when all diodes are ON (as shown in Figure 9a). For the entire frequency range, a nearly stable response is achieved. Moreover, different diode formations lead to a quantitative feature in the phase shifter. Only phase shift performance in the 4.5 to 7 GHz frequency range is shown in the charts.

Ideally, phase shifters should provide a constant phase shift and dominant signal transmission. The signal coupling parameter in Figure 8 shows high quality wave transmission. In addition, a nearly constant phase difference is achieved based on the analysis of Figure 9, considering every frequency point.

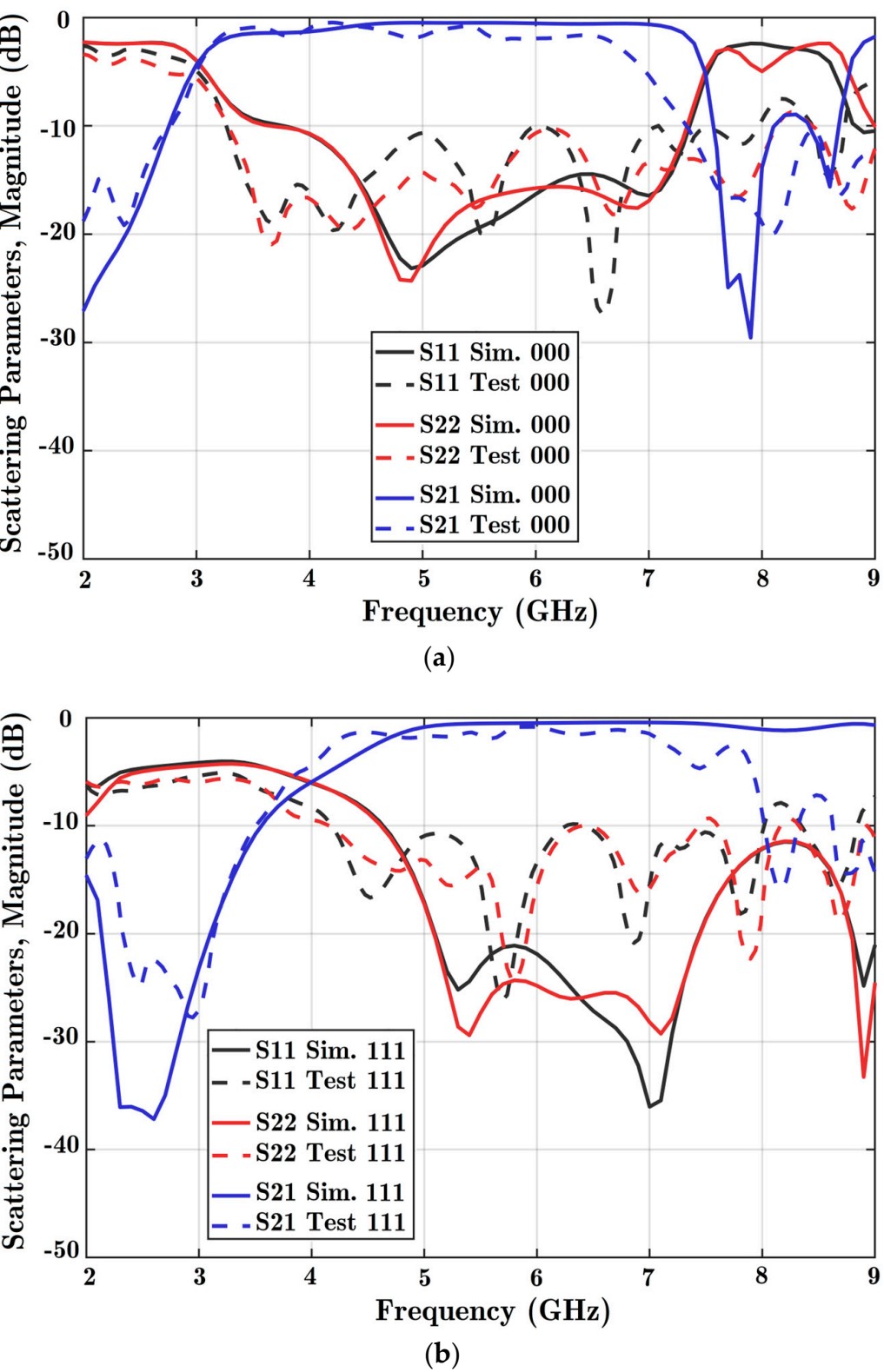

**Figure 8.** Scattering parameters versus frequency for two states: (**a**) all diodes OFF, and (**b**) all diodes ON. Reflection coefficients for port 1 (S11), port 2 (S22), and the coupling parameter between the ports (S21) are obtained. Simulation software charts are in solid colors, and the experiment is shown in dashed lines. State 000 is the situation with all diodes OFF, while state 111 is three diodes ON at the same time.

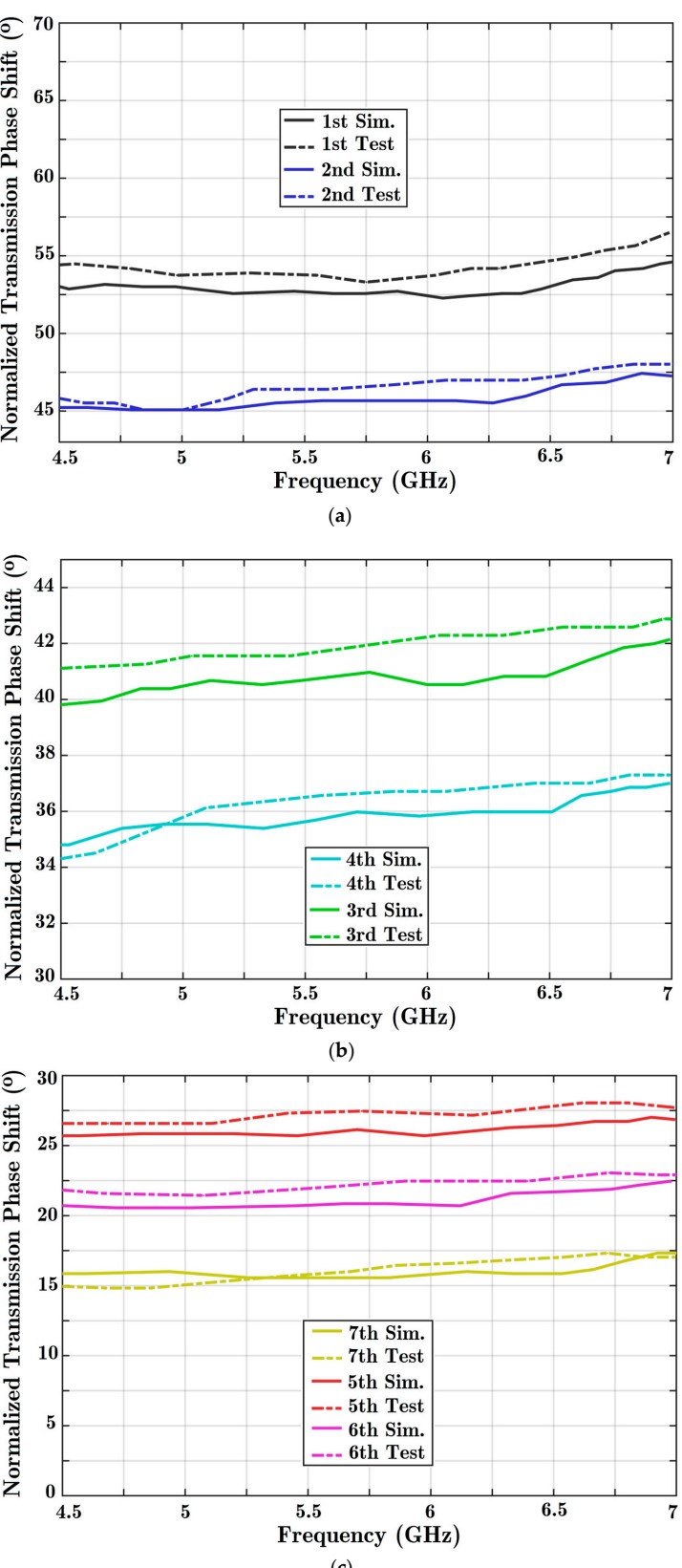

**Figure 9.** The achieved phase shift quantities are presented using the proposed reconfigurable structure with three parallel diode posts integrated with a HM-SIW transmission line. Comparisons are made between the results of two concrete simulations and experiment solutions: (**a**) the first phase range of 45° to 70°; (**b**) the medium phase range of 30° to 45°; and (**c**) the primary phase steps of 0° to 30°.

In order to compare the achieved results with similar phase shifter designs in the literature, Table 1 was prepared. The proposed structure has a promising configuration compared to recent phase shifter designs in the literature. In addition to the tunable states (seven normalized modes), the DC biasing of PIN diodes is integrated into a similar layer as the RF signal. This feature eliminates the need for an extra bias layer (preventing multilayer designs). Furthermore, the half-mode design decreases the width of the SIW phase shifter, which makes it an ideal device for modular and integrated systems. Miniaturized size and enhanced DC biasing technology enable a reliable, cost-effective electronic component that can be integrated into a package. As an example, the proposed system can be applied to a sensing antenna for the phase engineering section. Beam steering systems require compact phase shifters. With a phase shifter, it is possible to realize a high-resolution detective system for applications such as automatic radar sensors. Another application of this innovative phase shifter is for time-varying surfaces.

**Table 1.** A comparison table with previously reported SIW phase shifter works and their performances.

| References | [20] | [24] | [25] | [26] | [27] | [28] | [29] | [30] | [35] | This Work |
|---|---|---|---|---|---|---|---|---|---|---|
| Technology | Two reconfigurable filter couplers | Change in the diameter and position of posts | Change length of slots | Two slots location | Control the distance of a high dielectric constant | Different cross-sectional elements | Change in the position of posts | Change the waveguide loading | Tunable resonators with varactor | Parallel embedded component with HM-SIW |
| Frequency (GHz) | 1–1.3 | 10 | 10–14 | 26 | 29–31 | 4.2–7.2 | 22.5–26.5 | 9.5–10.5 | 6.5 | 4.5–7 |
| Phase difference (°) | 300 | 55 | 90 and 180 | −90, −45, 0, 45, 90 | 275 | 45 and 90 | 60 | 45 | 360 | ~60 |
| Insertion loss (dB) | −4 | −1 | −1.5 | −1.5 | −2 | −2.5 | −1.5 | −1 | −3.5 | <−2 |
| Number of layers and complexity | Complex | 1 | 2 | 1 | Complex | 1 | 1 | 2 | 1 | 1 |
| Reconfigurable | √ | × | × | × | √ | × | × | √ | √ | √ |
| Size ($\lambda_0 \times \lambda_0$) | 1.1 × 0.9 | 0.84 × 0.33 | × | 1.9 × 1.43 | 0.6 × 0.4 | 0.8 × 0.67 | 3.6 × 0.7 | 1.5 × 0.5 | 0.91 × 0.3 | 0.76 × 0.31 |

√ The structure is benefit from this feature. × There is no information on the required feature.

Furthermore, a revised model is simulated using simulation software in Appendix A to illustrate the capabilities of the proposed HM-SIW structure. It is possible to cascade two or more phase shifters to generate higher phase shift values. Signal loss happens at port connections and transitions, so cascading more phase shifter components does not generate more loss. This expanded design is included in Appendix A of the simulation software. The total size of the redesigned model will increase by 41%.

Moreover, a parametric study is provided in Appendix B. The parametric study covers the nonlinearity and variation of PIN diode characteristics, including the equivalent parallel circuit models. In this case, the diode parameters such as resistance, inductance, and capacitance were tuned to further investigate their effects on the scattering parameters of the HM-SIW. As well as using full wave simulation software to analyze the compact structure, circuit simulation software is used to study the different circuit elements modeling the diodes.

## 4. Conclusions

A brand-new half-mode substrate integrated waveguide (HM-SIW) phase shifter is proposed here to cover 4.5 to 7 GHz frequencies. The structure is based on microstrip lines integrated into HM-SIW, PIN diodes, and radial stubs. It is optimized to achieve low insertion (<−2 dB) and reflection losses. By exploiting a half-mode SIW formation, it results in a smaller dimension than a typical SIW formation. A parametric study achieves high performance with multiple phase shifts, scattering parameters, and a miniaturized modular design. The primary design concept covers 0° to 60° phase shifts with 7 distinct states.

By cascading two similar topologies without extra wave dissipation, the phase shift can reach 120°. Furthermore, the observed phase deviation was less than 5° in the frequency band. The structure is innovatively designed on one substrate layer. This makes it compact, low-loss, and low-profile, so it can be easily combined with other modular microwave devices. It does not require extra wire binding for DC biasing the tunable element (PIN diode). This makes it attractive for millimeter-wave frequencies and applications such as sensing and antenna beamforming.

**Author Contributions:** Conceptualization, H.B. and F.D.W.; methodology, H.B. and F.D.W.; software, A.G., B.M. and F.D.W.; validation, H.B., A.G. and F.D.W.; resources, H.B. and L.T.; writing—original draft preparation, B.M. and F.D.W.; writing—review and editing, H.B. and A.G.; supervision, H.B. and L.T. All authors have read and agreed to the published version of the manuscript.

**Funding:** This research received no external funding.

**Informed Consent Statement:** Not applicable.

**Data Availability Statement:** The data presented in this study are openly available.

**Conflicts of Interest:** The authors declare no conflict of interest.

## Abbreviations

In order to help readers, some key terms and symbols are described in this part as follows.

| | |
|---|---|
| SIW | Substrate integrated waveguide |
| HM-SIW | Half-mode substrate integrated waveguide |
| RF | Radio frequency |
| RLC | Resistor, inductance, capacitance |
| SOLT | Short, open, load, through |

CST, ANSYS HFSS, AWR, and ADS are different simulation software

## Appendix A

It should be noted that all diode states can prepare for a phase shift from 15° to 55°. More phase shift is possible by cascading the structures without extra loss in the wave transition. The main dissipation in the signal is from the microstrip to SIW transition, including the port matching mechanism. This appendix provides a cascaded structure similar to the proposed design. For this study, simulation software is used only to elaborate on the capabilities of a HM-SIW phase shifter. The updated structure has a length of $L_{new}$ = 73 mm, as illustrated in Figure A1.

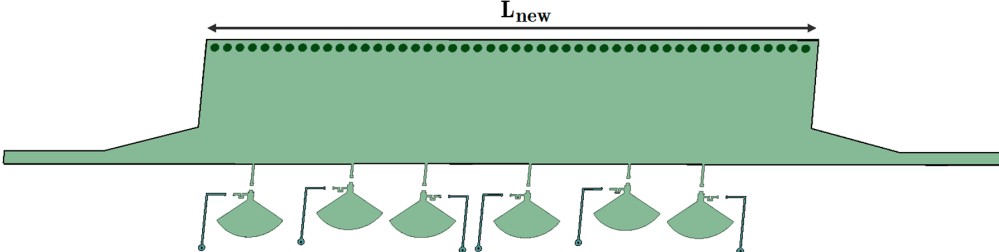

**Figure A1.** A proposal for cascading the structure of the phase shifter with the HM-SIW to generate twice the primary phase shifts.

It is achieved that the operating frequency of this cascaded phase shifter is 4.8 to 7.4 GHz for the extended design. It is the result of a simple transforming property of the design of the structure without additional geometric optimization in the simulation medium. In CST software, only a copy-transfer operation is utilized to create a simple extended feature for this purpose. With proper tuning of the cascaded structure, it is possible to achieve a similar frequency band. Scattering parameters are shown in Figure A2 for both reflection and coupling coefficients. Only two scenarios, all diodes OFF and all

diodes ON, are considered here for the sake of brevity. Within the assigned frequency bandwidth, the coupling value was less than −2 dB, and the reflection coefficient of the two ports was less than −10 dB.

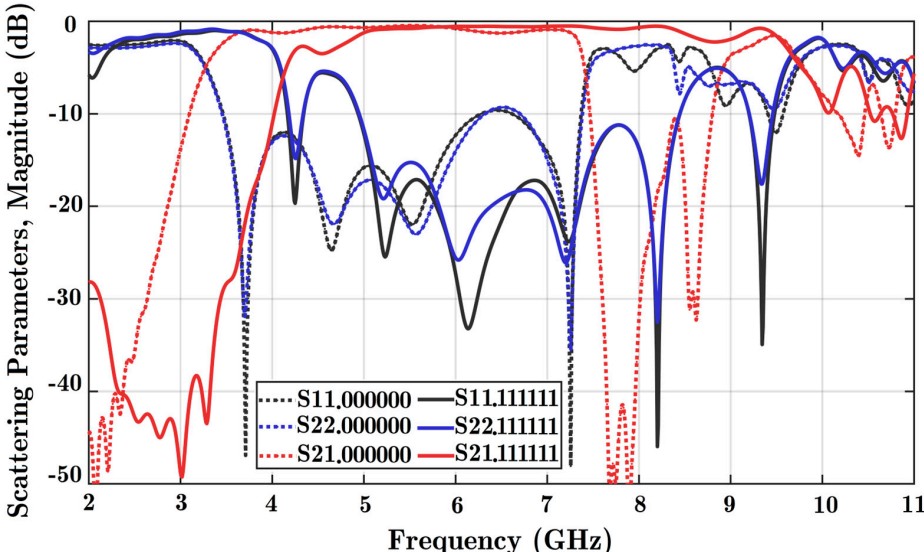

**Figure A2.** Scattering parameters versus frequency for two states: all diodes OFF, and all diodes ON. Reflection coefficients for port 1 (S11), port 2 (S22), and the coupling parameter between the ports (S21) are obtained. Only simulation software is executed for this cascaded phase shifter element. State 000000 is the situation with all diodes OFF (dotted lines), while 111111 represents six diodes ON at the same time (solid lines).

A similar diode feature is added to the design. Moreover, the number of diodes has increased to six in this extended design. The purpose of a cascaded phase shifter based on the proposed HM-SIW is to show the capability of the microwave device to achieve higher phase shifts. The increased number of diodes also leads to more phase shifts. As shown in Figure A3, the extended structure achieved the maximum value of phase shift. Now, it is possible to have up to 120° of phase shift. By cascading three phase shifters, the ultimate transmission phase shift value would be 180°.

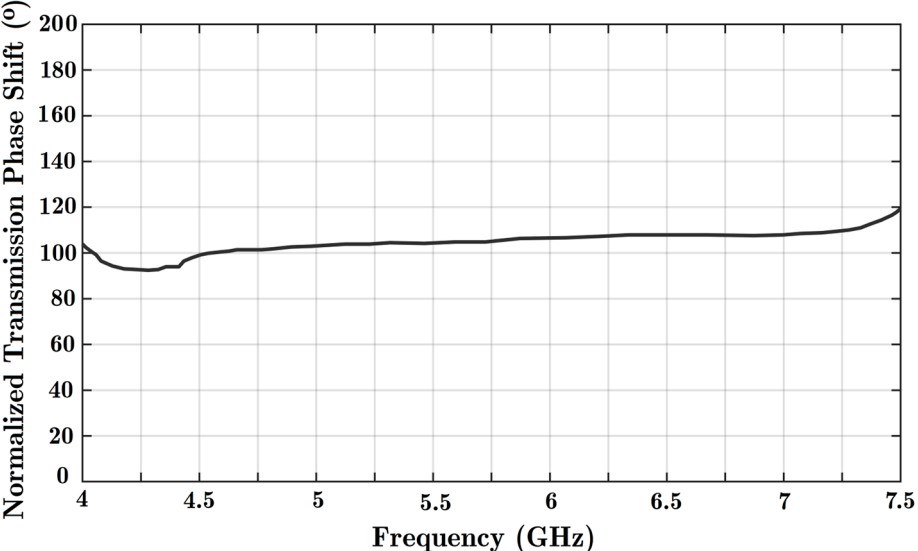

**Figure A3.** The maximum achieved phase shift value is presented using an extended reconfigurable structure with six diode posts integrated with a HM-SIW transmission line.

## Appendix B

As PIN diodes are nonlinear elements, their equivalent circuit models can be varied by altering the supply voltage and bias source [47]. It is possible to use a precise DC source and a fast switch key to change the state of a PIN diode with more consistency. As another option, the proposed phase shifter can be applied in a receiving section using a lower amount of RF power signal (a small signal). This will improve the stability of the diode.

A parametric analysis was performed here with the purpose of applying a 20% variation to some equivalent circuits of the applied PIN diode in simulation software. The components to be studied are the resistor and inductance values in the ON state and the capacitance in the OFF state ($C_{off}$ = 0.10704, 0.1338, and 0.16056 PF).

To perform parametric studies, it is beneficial to use circuit software, such as ADS or AWR, rather than working with full-wave CST or ANSYS HFSS. This is because circuit software is here to tune equivalent circuit elements. First, the structure is simulated without PIN diodes in CST. To replace the diode component, discrete ports are added that connect to the ground layer. Then, there would be eight different ports, including the two main input and output ports in the case of the proposed fabricated structure. In the next step, scattering parameters can be exported in s8p format with a normalized 50 Ω impedance. Then a s8p file can be applied to an 8-port scattering module in ADS software. Finally, it is an easy and quick way to tune the equivalent parameters of the diode in circuit software to achieve the desired scattering parameters.

Figure A4 is the parametric study for the OFF state on the scattering parameters. Similarly, Figure A5 presents the variation of the OFF state capacitance in the transmission phase. Changing the ON state parameters does not affect either the magnitude or phase of the scattering parameters in a sensible way.

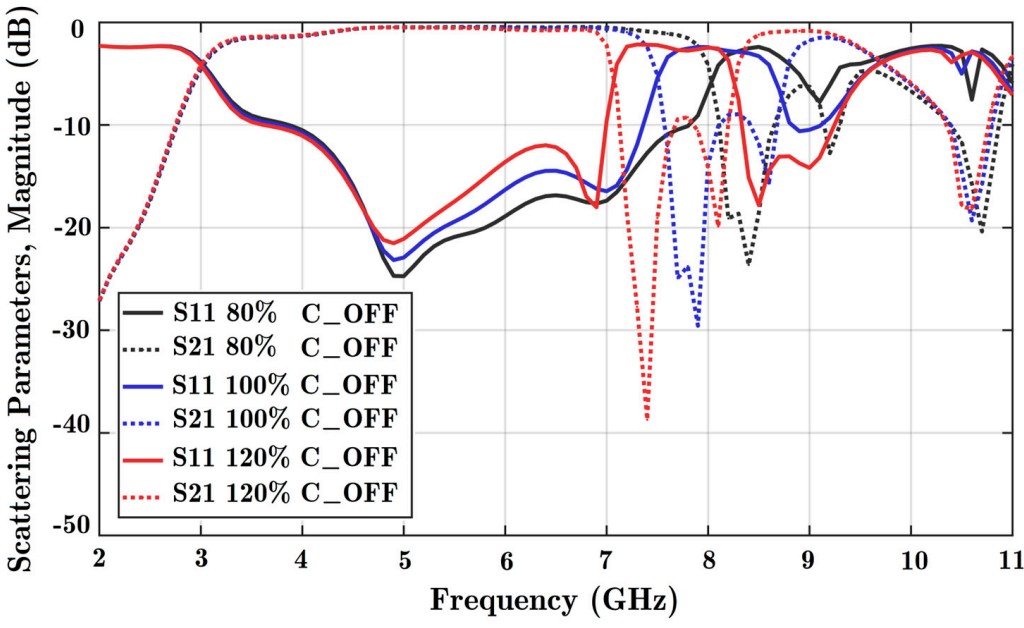

**Figure A4.** Scattering parameters versus frequency for parametric studies when all diodes are OFF. Reflection coefficients for port 1 (S11) and the coupling parameter between the ports (S21) are obtained. Only simulation software was used for these parametric studies. Dotted lines indicate coupling, while solid lines represent reflection coefficients (80% C = 0.10704, 100% C = 0.1338, and 120% C = 0.16056 PF).

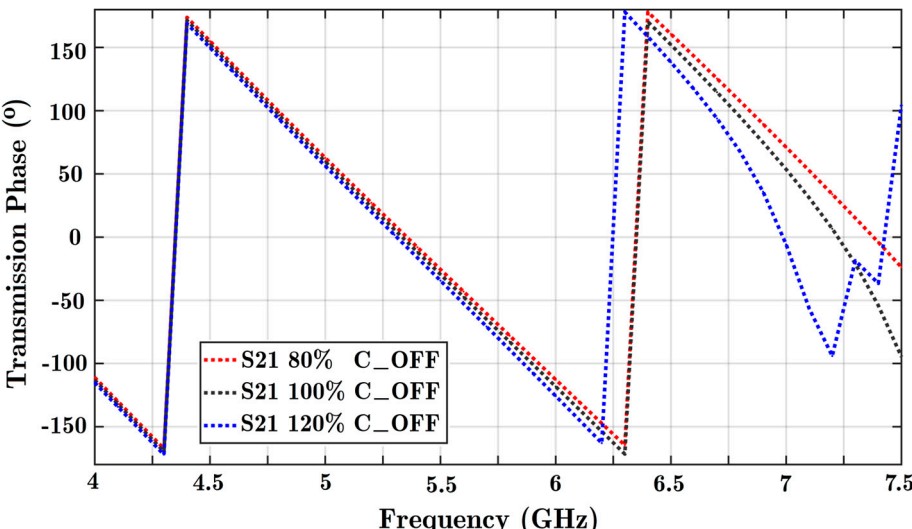

**Figure A5.** Variation of the transmission phase using different capacitance parameters in the off state. Changes in the transmission phase are noticeable at higher frequencies. A frequency range of 4 to 7.5 GHz is considered here.

The parametric analysis shows that with a lower-capacitance PIN diode, it is possible to increase the operating frequency band of the phase shifter.

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
