# Peer review of "Miniaturized Compact Reconfigurable Half-Mode SIW Phase Shifter with PIN Diodes"

_technologies, doi:10.3390/technologies11030063_

Round 1

Reviewer 1 Report

The manuscript entitled “Miniaturized Compact Reconfigurable Half-Mode SIW Phase Shifter with PIN Diodes” proposes a half-mode SIW-based phase shifter, which is reconfigurable thanks to the activation/deactivation of 3 diodes, thus yielding to 8 different configurations. The methods are sound and the results are good, but the scientific contribution is not fully clear. Also, it is difficult to see how this manuscript could fall into the journal’s scope. The topics covered by the manuscript seem more suitable for journals such as IEEE Trans. Microw. Theory Tehn., IEEE Microw. Wirel. Comp. Lett., Microw. Opt. Terchnol. Lett., or Electronics (MDPI), among others. The lack of a clear significant contribution and the lack of suitability of this manuscript as for the journal’s scope makes me advise to reject the manuscript. The authors might want to revise it and submit it to a more suitable journal. Please find below some extra comments:

1.     The novelty or real contribution is a little bit arguable: in comparison with other state-of-the-art phase shifters, this one only features a certain size reduction. While interesting, this might not be enough breakthrough for a formal scientific publication. This feature needs to be better justified: why this size reduction is so critical nowadays? Which are the current problems modern phase shifters are facing

2.     According to Table 1, there is no information about the “miniaturized” feature for any of the references under comparison, thus again making the contribution of this new work to seem weak. A comprehensive size analysis would be desirable.

3.     It is not clear how some of the parameters for the transmission line (Sect. 2.1) were selected, such as Wg, d or p.

Author Response

1. Compared to existing phase shifter designs, the proposed structure has some promising configurations. (1) DC biasing of PIN diodes is integrated into a similar layer to the RF signal. This shows that an innovative technology is exploited for phase shifters. This feature eliminates the need for an extra bias layer (preventing multilayer designs). (2) Furthermore, the half-mode design decreases the width of the SIW phase shifter, which makes it an ideal device for modular and integrated systems. (3) Therefore, the miniaturized size and enhanced DC biasing technology enable a reliable, cost-effective electronic component that can be integrated into a package system.

2. Yes, this section of Table 1 has been modified to present the miniaturization process in numbers. As compared, the proposed work shows an acceptable improvement.

3. Equations (1) and (2) are applied to calculate the transmission line geometry for 4 GHz when modeling SIW with a dielectric-filled rectangular waveguide. Then, the Weff is split in half for a half-mode SIW and Wg=14.5 mm. Furthermore, the diameter of the via (d=1 mm) and the distance between the vias (p=1.5 mm) are set up with equations (3) - (9) and the fabrication limits.

Reviewer 2 Report

Paper is well written, impressed with the work. The only recommendation for authors is to add a table of nomenclature or symbols, which will help readers easily relate to the important terms and symbols used in the article.

Author Response

Thank you very much. In order to help readers, some key terms and symbols are described in this part as follows.
SIW: Substrate integrated waveguide
HM-SIW: Half-mode substrate integrated waveguide
RF: Radio frequency
RLC: Resistor, inductance, capacitance
SOLT: Short, open, load, through
CST, ANSYS HFSS, AWR, and ADS are different simulation software

Reviewer 3 Report

The manuscript presents a reconfigurable Half-Mode SIW Phase Shifter using PIN Diodes. The investigation is sufficient and the content is well organized. I have some concerns:

1)     In the proposed design, reconfigurable is reflected in seven discontinuous states. Can this design realize continuously reconfigurable by using varactor.

2)     How to choose the added position of the PIN diodes. Additionally, the design procedure is suggested to be added.

3)     The in-band phase deviation is suggested to give in the article.

4)     Table 1, it is recommended to add a column of sizes of the designs to prove that the proposed design is miniaturized.

Author Response

1. Yes, varactor diodes in theory can provide various continuous features. The response of PIN diodes, however, is concrete and stable regardless of the variation in the input source. In view of practical considerations, a PIN diode is used here to intervene in the transmission line.

2. When looking at Figure 2, the added length of lines connecting to the PIN diodes presents a constant phase shift for every branch. Additionally, they provide enough space for a diode soldering pad. The location of PIN diode stubs is tuned with full wave simulation considering fabrication constraints and achieving a sensible phase shift for a smaller total length. With that being said, it is possible to choose a shorter distance between diode stubs if practical considerations are ignored. When diodes are placed closer together, the transmission line between two parallel posts is smaller, so the operating frequency bandwidth is less. It is based on the equivalent model of a half mode SIW to a rectangular dielectric filled waveguide.

3. The observed phase deviation was less than 5° in the frequency band. It is added to the conclusion part.

4. Yes, this section of Table 1 has been modified to present the miniaturization process in numbers. As compared, the proposed work shows an acceptable improvement.

Reviewer 4 Report

A comparison table can be included with some state-of-art similar works, so that to show the advantages and innovation of the proposed tunable phase shifter.

Author Response

Yes, Table 1 has been modified to include more information on the size of similar and different phase shifter technologies.

Reviewer 5 Report

This work presents an electrically reconfigurable phase shifter based on a half-mode substrate-integrated waveguide (HM-SIW). The proposed design uses PIN diodes for tunability and is a single-layer structure that does not require any extra wiring layer for the bias circuit. The proposed half-mode structure is around half the size of a typical SIW line, and with the proposed design, the seven states of the PIN diodes can be validated with a relatively constant phase difference across a broad frequency range.

Sadly, reading the manuscript is a huge letdown and provides the reader with almost little new information. There is little novelty in the work.  Already similar works have been published.

There are too many self-citations. References are given below:

10,11,22,23,31,32,33,36, 3,5,6,7, 42

Its current version does not meet publication criteria. The introduction does not provide a thorough overview of the subject, and the number of references is insufficient. The conclusion section of the paper is far too brief. The manuscript requires extensive revision.

Author Response

1. Compared to existing phase shifter designs, the proposed structure has some promising configurations. (1) DC biasing of PIN diodes is integrated into a similar layer to the RF signal. This shows that an innovative technology is exploited for phase shifters. This feature eliminates the need for an extra bias layer (preventing multilayer designs). (2) Furthermore, the half-mode design decreases the width of the SIW phase shifter, which makes it an ideal device for modular and integrated systems. (3) Therefore, the miniaturized size and enhanced DC biasing technology enable a reliable, cost-effective electronic component that can be integrated into a package system.

2. In order to simplified the text, some of the references and their related parts are shorted in the modified version.

3. We have reviewed and revised the abstract, introduction, and conclusion sections of the paper.

Round 2

Reviewer 1 Report

The main improvement featured by this device is the miniaturization, it has no other extraordinary charachteristic. However, looking at Table 1 there are previous devices showing quite similar miniaturizations (such as [24], [27] or [35], for example). The contribution is, once again, too weak for a formal scientific publication. The manuscript is well written, and this is doubtlessly a good technical work, but it does not seem a good scientific work. What can we learn from this work? Which is the scientific advancement? These issues, in addition to the non-suitability of this manuscript for this journal (again, it would be more suitable to submit to Electronics, IEEE Trans. Microw. Theory Techn., IEEE Microw. Wirel. Comp. Lett. or other similar journals), makes me advice to reject the manuscript.

Reviewer 5 Report

The authors have addressed the comments.